# Management outcomes of secondary glaucoma due to retinopathy of prematurity: A 19-year prospective study at a tertiary eye care Institute. The Indian Twin cities ROP Screening (ITCROPS) database report number 8

Sirisha Senthil[1,2]*, Pasyanthi Balijepalli[1], Ashik Mohamed[3], Padmaja Kumari Rani[4], Sameera Nayak[5], Chandrasekar Garudadri[1], Anil K. Mandal[1,2], Subhadra Jalali[2,4]

1 VST Glaucoma Center, L V Prasad Eye Institute, Hyderabad, India, 2 Jasti V Ramanamma Children's Eye Care Center, L V Prasad Eye Institute, Hyderabad, India, 3 Ophthalmic Biophysics, L V Prasad Eye Institute, Hyderabad, Telangana, India, 4 Srimathi Kanuri Santhamma Centre for Vitreoretinal diseases, L V Prasad Eye Institute, Hyderabad, India, 5 Kode Venkatadri Chowdary Campus, Vitreoretinal Services, L V Prasad Eye Institute, Vijayawada, India

* sirishasenthil@lvpei.org

## Abstract

### Purpose

To report the clinical presentation and management outcomes of glaucoma in the "Indian Twin cities retinopathy of prematurity (ROP) Screening database."

### Methods

All children with diagnosis of ROP and glaucoma between 1997 and 2016 from a prospective database were included. Glaucoma was classified as open when anterior chamber (AC) was deep, closed when AC was shallow or flat and neovascular when there was extensive iris neovascularization. ROP was classified based on International classification of ROP.

### Results

The prevalence of secondary glaucoma in our cohort was 1.36% (82 eyes of 6000 children). Eighty-two eyes of 54 children with secondary glaucoma due to ROP where included in this study. The distribution of glaucoma among the ROP stages included, stage V (58.5%), stage 1V (24.3%), stage III (2.4%) and stage II (1.2%) eyes. Median (interquartile range) duration from birth to glaucoma diagnosis was 7.8 (4.2, 24.9) months. Type of glaucoma was angle closure in 39 (47.6%), open angle in 35 (42.7%) and neovascular in 8 (9.8%) eyes. Retinal interventions included vitreoretinal surgery in 59 (72%), retinal laser in 14 (17%) and intravitreal bevacizumab injection in 19 (23.1%) eyes. The mean (±standard deviation) IOP at presentation was 22.6 ±11.8 mm Hg. Glaucoma was managed medically in 66

**Data Availability Statement:** Data cannot be shared publicly because of confidentiality. Data is available from the corresponding author (sirishasenthil@lvpei.org) or the institutional review board of L V Prasad Eye Institute (irb@lvpei.org). Other researchers will be able to access the data in the same manner as the authors. The authors did not have any special access privileges.

**Funding:** This work was supported by the Hyderabad Eye Research Foundation. The funder had no role in study design, data collection and analysis, decision to publish, or preparation of the manuscript.

**Competing interests:** The authors have declared that no competing interests exist.

(76%) and surgically in 16 (19.5%) eyes. The mean follow up for the entire cohort was 1.14 ±2.24 years. At final visit, 37% eyes with ROP and glaucoma had ambulatory vision with mean IOP of 16.0±8.1 mm Hg and 56 eyes (68.2%) needed glaucoma medications.

## Conclusion

In this large ROP cohort, 1.36% eyes developed secondary glaucoma. Majority of them had stage V or IV ROP and 1/5 of them needed glaucoma surgery. Around 1/3rd of the ROP eyes with glaucoma had ambulatory vision.

## Introduction

Retinopathy of prematurity (ROP) is a disease of developing retinal vasculature in premature infants. It is a disease of modern era where application of technology to save the lives of these premature children is associated with sight threatening consequences. The first ever cases have been reported in 1940 [1] and is estimated that by 2020, ROP will be the single largest cause of potentially avoidable blindness in children [2]. Various retinal and non-retinal complications have been reported in children with ROP and these complications increase with increasing severity of ROP [3].

Glaucoma is an important sight threatening complication among these children particularly with severe stages of ROP, with nearly 1/3 of all eyes of unoperated stage V ROP developing glaucoma [4, 5]. Elevated intraocular pressure (IOP) can occur secondary to advanced ROP itself or following surgical treatment for ROP [5, 6]. Various mechanisms for the development of glaucoma in eyes with ROP have been proposed. These include, forward displacement of lens iris diaphragm with progressive or acute angle closure in eyes with cicatricial ROP [7, 8], neovascular glaucoma, glaucoma following vitreoretinal (VR) surgery, cilio-lenticular block glaucoma [9] and arrest or delay in angle development [10]. Laser for ROP can rarely be associated with recurrent hyphema or elevated IOP secondary to angle closure, which may resolve without long-term sequelae or may need intervention [11, 12].

The prevalence of secondary glaucoma in children with ROP range from 2% to 30%, with highest prevalence in eyes with advanced untreated ROP [5, 13, 14]. In the Early Treatment for ROP (ETROP) study, nearly 2% of eyes with high-risk pre-threshold ROP developed glaucoma [13]. In a study by Chandra et al, following VR surgery in eyes with stage IV ROP, glaucoma was noted in 6% of eyes [6]. Glaucoma in ROP is reported to have poor prognosis for vision [4, 13]. Elevated IOP can occur in the early or late postoperative period following VR surgery and significant proportion of them with delayed presentation need surgery for IOP control [5].

All publications are limited by small case series or individual case reports. Our study objective was to present the incidence, causes, treatment options and outcomes of glaucoma in premature infants with various stages of ROP from a large prospective database.

## Materials and methods

As a part of ongoing prospective study "The Indian Twin Cities Retinopathy of prematurity Screening (ITCROPS) program database", we evaluated the children with ROP and glaucoma. We included children evaluated between January 1, 1997 to May 31st, 2016. Details of screening, enrolment and treatment protocols have been described elsewhere [15]. The institutional

review board of L V Prasad Eye Institute and Hyderabad Eye Research Foundation have approved this prospective database (LEC 08088). Informed consent was obtained from the parents of the children and the study adhered to the tenets of the Declaration of Helsinki. Trained retina specialists evaluated the babies and the children were referred to pediatric glaucoma specialists when further management was required. The data collected included, demographic data, birth and neonatal history, birth weight, gestational age at birth, history of incubation and oxygen inhalation. Details of blood transfusion and septicaemia were evaluated. Clinical details like corneal diameter, clarity, scar, anterior chamber depth, peripheral anterior synechiae, neovascularisation of iris, IOP at presentation and at subsequent follow-ups, cup-disc-ratio and retinal status were collected. Details of retinal evaluation including staging of ROP, type of intervention, Laser indirect ophthalmoscopy (LIO), intravitreal bevacizumab and VR surgery were collected. Need for antiglaucoma medications at presentation and at subsequent follow-ups, need for glaucoma surgery and their outcomes were reviewed.

ROP was classified according to International classification of Retinopathy of prematurity (ICROP) from Stage 1 to Stage V, based on the abnormal vascular response at the junction of vascularised and avascular retina [16]. Preterm babies ($\leq$35 weeks gestational age and $\leq$2000 gm birth weight) were included in the study.

We defined glaucoma when two or more of the following parameters were present [17]; 1) IOP >16 mm Hg under anesthesia (or $\geq$21 mm Hg in the clinic) recorded with Perkins tonometer or elevated IOP by digital assessment, 2) presence of corneal edema/Haab's striae/ corneal diameter $\geq$12 mm within 1 year of birth), 3) presence of glaucomatous disc damage: cupping of the disc/rim thinning, 4) progressive myopia with increase in ocular dimensions beyond the normal growth for age in these children with ROP, 5) shallow/flat anterior chamber (AC) with or without iris neovascularization. Examination under anesthesia was done for evaluation and follow up as and when deemed required.

The glaucoma in eyes with ROP was classified into congenital and secondary glaucomas based on the presentation [18]. Diagnosis of congenital glaucoma was based on elevated IOP at birth with corneal changes suggestive of congenital glaucoma (corneal edema, Haabs' striae, enlarged corneal diameter) prior to any intervention for ROP. It was classified as secondary when the above ocular characteristics were present with either a prior history of vitreo-retinal surgery or with features suggestive of angle closure (closed angles, shallow AC, neovascularization). Open angle glaucoma was diagnosed when the AC was deep or open angles on gonioscopy, angle closure glaucoma when the AC was shallow or angles closed on gonioscopy, and neovascular glaucoma when there was iris neovascularisation. When IOP was high, topical antiglaucoma medications were prescribed in the order of dorzolamide, timolol maleate followed by prostaglandin analogue. If the IOP remained persistently high or corneal edema persisted or if there were other signs of progression like the enlargement in corneal diameter/ Habb's striae or increase in disc cupping, surgical intervention was planned and performed. Our choice of glaucoma surgery and the technique of combined trabeculotomy and trabeculectomy are as described in our earlier publication [18].

Primary outcome measure was IOP control and secondary outcome measure was visual outcomes. Outcome with retinal intervention was termed 'good' when retina was flat at centre and periphery, fair when macula was attached and poor when macula was detached.

## Statistical analysis

The statistical analysis was performed using the software Origin v7.0 (Origin Lab Corporation, Northampton, MA, USA). Continuous data were checked for normality using Shapiro-Wilk test. Descriptive statistics included mean and standard deviation (SD) for normally distributed

variables and median with inter-quartile range (IQR) for non-parametric data. Categorical variables were summarized as proportions. Comparison among various stages of ROP (IV-A, IV-B and V) was performed using Kruskal-Wallis test for non-parametric continuous data and Chi-square test for categorical data. A p-value of <0.05 was considered statistically significant. Multiple pair-wise comparison was performed by Mann-Whitney test for non-parametric continuous data and Fisher Exact test for categorical data, and a p-value of <0.025 was considered statistically significant after adjustment for Bonferroni correction.

## Results

Of the 15000 preterm babies evaluated during the study period, 3000 children were diagnosed with ROP. Of these, 87 eyes of 57 children were diagnosed with glaucoma. Five eyes of 3 children had primary congenital glaucoma with ROP, the details of which are already published [18]. Excluding the 5 eyes with Primary Congenital Glaucoma (PCG), we included 82 eyes of 54 children with ROP and secondary glaucoma in this study. Median age at first presentation of these preterm children was 3.9 (IQR, 2.3–8.8) months. Median duration from birth to glaucoma diagnosis was 7.8 months (IQR, 4.2–24.9 months). The mean follow up was 1.14±2.24 years and range of follow-up was 1 day to 12.6 years.

### Demographic features

There were 28 boys and 26 girls. Mean birth weight of these children with ROP and glaucoma was 1200 ± 300 grams. Mean (95% confidence limits) gestational age at birth was 29 (28,31) weeks. Comparison of clinical characteristics at presentation in various stages of ROP is shown in Table 1. The gestational age at birth, the birth weight, gender were comparable in all stages of ROP. However, the median age at presentation was more for stage V ROP (4.9 (IQR, 3.2–13.8) months) compared to stage IV-B (p = 0.009).

### Clinical features

Corneal edema was noted in 23 eyes (28%) of which 6 belonged to stage V ROP, Haab'striae in 4 eyes (4.9%) and enlarged corneal diameter in 6 eyes (7.3%). Corneal opacity/scar was seen in 5 eyes (10.4%) all belonging to stage V ROP. Anterior chamber was shallow in 39 eyes (47.6%), deep in 35 eyes (42.7), neovascularisation of the iris was noted in 8 eyes (9.8%). The mean

**Table 1. Demographic and clinical features in various stages of ROP at presentation.**

| Characteristic | | ROP Stage II and III* | ROP Stage IV-A | ROP Stage IV-B | ROP Stage V | p-value |
|---|---|---|---|---|---|---|
| | | (n = 3 eyes; 3 patients) | (n = 11 eyes; 8 patients) | (n = 20 eyes; 13 patients) | (n = 48 eyes; 34 patients) | |
| **Age at presentation (in months), median (IQR)** | | 0.8 (0.8 to 7) | 2.4 (1.3 to 3.8) | 2.8 (2.2 to 4.1) | 4.9 (3.2 to 13.8) | **0.02**[a] |
| **Gender** | Male | 1 (33.3%) | 4 (50%) | 6 (46.2%) | 19 (55.9%) | 0.83 |
| | Female | 2 (66.7%) | 4 (50%) | 7 (53.8%) | 15 (44.1%) | |
| **Birth weight (g), mean ± SD** | | 1433.3 ± 57.7 | 1115.7 ± 201.0 | 1144.2 ± 209.6 | 1259.7 ± 361.2 | 0.39 |
| **Gestational age at birth (weeks), median (IQR)** | | 36 (28 to 36) | 28 (28 to 29) | 28.5 (28 to 30) | 30 (28 to 31) | 0.26 |
| **Cornea** | Clear | 2 (66.7%) | 10 (90.9%) | 18 (90%) | 37 (77.1%) | 0.32 |
| | Edema or Opacity | 1 (33.3%) | 1 (9.1%) | 2 (10%) | 11 (22.9%) | |

IQR: interquartile range; SD: standard deviation;

*Not included in analysis because of small sample size;

[a]Kruskal-Wallis test; Post-hoc analyses (multiple comparisons by Mann-Whitney test) showed that only stage IV-B was significantly different from stage V (p = 0.009).

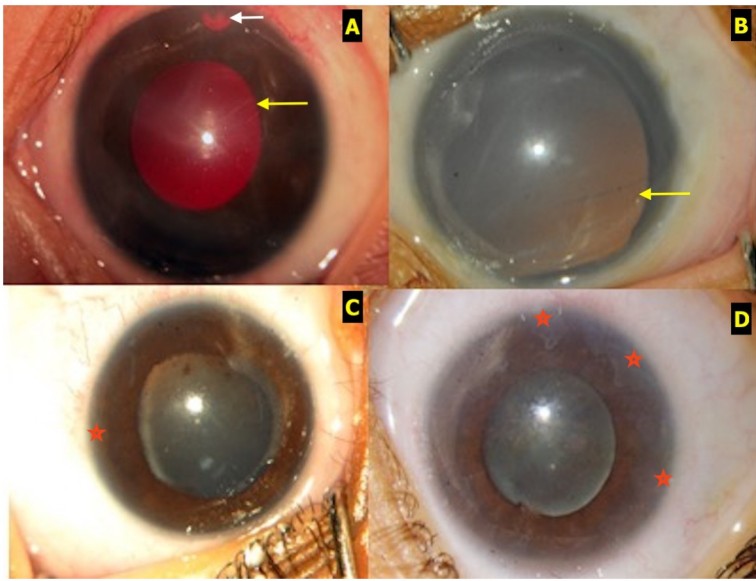

**Fig 1. A**: Anterior segment photograph showing an eye with ROP and secondary glaucoma that underwent combined trabeculotomy with trabeculectomy. Note the patent surgical PI (white arrow), clear cornea and Habb's striae (yellow arrow). **B**: Anterior segment photograph of an eye with ROP post vitreoretinal surgery, aphakia, corneal haze, dilated pupil with ectropion uveae and Habb's striae (yellow arrow). **C**: Anterior segment photograph of an eye with ROP post-lens sparing vitreoretinal surgery with dilated fixed pupil, posterior synechiae and few peripheral anterior synechiae (red star). **D**: Anterior segment photograph of an eye with ROP and angle closure glaucoma with extensive peripheral anterior synechiae (red stars).

(±SD) IOP at presentation was 22.3±11.3 mmHg. Fig 1 shows a few examples of the ROP eyes with glaucoma.

## Treatment for ROP

In this cohort, retinal laser with indirect ophthalmoscopy (LIO) was performed in 14 eyes (17%), and intravitreal bevacizumab injection was given in 19 eyes (23%); (16 eyes received one injection and 3 eyes received 2 injections). The dose of intravitreal bevacizumab used was 0.62 mg in 0.025 ml. A total of 59/82 eyes (72%) underwent VR surgery and 8 eyes underwent retinal surgery twice. Among the 59 eyes with VR surgery, 54 (91.5%) underwent concurrent lensectomy. Table 2 gives the details of ROP treatment in this cohort. Proportion of eyes that did not undergo any vitreoretinal surgery were comparable among stages IV-A, IV-B and V of ROP (p = 0.38). Age at first VR surgery was significantly more (p = 0.002) in stage V compared to stage IV-A. Greater proportion of eyes underwent retinal laser sittings in stages IV-A and IV-B respectively compared to stage V. The proportion of eyes that underwent intravitreal bevacizumab injection were significantly more in stage IVA compared to stage V (p = 0.005). Proportion of eyes that underwent simultaneous lensectomy along with VR surgery were comparable among stages IV-A, IV-B and V of ROP (p = 0.18).

**Types of glaucoma/ Association with glaucoma.** The most common stage of ROP associated with glaucoma was stage V ROP, seen in 48 eyes (58.5%), 20 eyes (24.4%) of Stage IVB, 11 eyes (13.4%) of Stage IVA, 2 eyes (2.4%) of Stage III and 1 eye (1.2%) of Stage II. No glaucoma was seen in Stage 1 ROP.

**Glaucoma diagnosis and treatment.** Median age at glaucoma diagnosis was 7.8 months (IQR, 4.2–24.9 months). Among the eyes with glaucoma, 57 eyes had elevated IOP of which 24 eyes had IOP ≥20 mm Hg with Perkins tonometer under general anesthesia and 33 eyes had

**Table 2. Retinal interventions and lensectomy in this cohort of ROP with glaucoma.**

| | | II and III* (n = 3) | IV-A (n = 11) | IV-B (n = 20) | V (n = 48) | p-value |
|---|---|---|---|---|---|---|
| **Vitreo-retinal surgeries** | No | 2 (66.7%) | 3 (27.2%) | 3 (15%) | 15 (31.2%) | 0.38 |
| | Yes | 1 (33.3%) | 8 (72.8%) | 17 (85%) | 33 (68.8%) | |
| **Age at first vitreo-retinal surgery (months), median (IQR)** | | N/A | 2.5 (1.8 to 3.6) | 3.4 (2.4 to 5.2) | 5.5 (3.8 to 7.3) | **0.004[a]** |
| **Laser indirect ophthalmoscopy** | No | 1 (33.3%) | 7 (63.6%) | 13 (65%) | 47 (97.9%) | **<0.0001[b]** |
| | Yes | 2 (66.7%) | 4 (36.4%) | 7 (35%) | 1 (2.1%) | |
| **Intravitreal Avastin** | No | 3 (100%) | 5 (45.45%) | 13 (65%) | 42 (87.5%) | **0.005[c]** |
| | Yes | 0 (0%) | 6 (54.55%) | 7 (35%) | 6 (12.5%) | |
| **Lensectomy** | | 0 (0%) | 7 (63.6%) | 17 (85%) | 30 (62.5%) | 0.18 |

IQR: inter-quartile range

*Not included in analysis because of small sample size

[a]Kruskal-Wallis test; Post-hoc analyses (multiple comparisons by Mann-Whitney test) showed that only stage IV-A was significantly different from stage V (p = 0.002)

[b]Chi-square test; Post-hoc analyses (multiple comparisons by Fisher Exact test) showed that only stage V was significantly different from stage IV-A (p = 0.003) and stage IV-B (p = 0.0005) and not stage IV-A vs stage IV-B (p = 1.00)

[c]Chi-square test; Post-hoc analyses (multiple comparisons by Fisher Exact test) showed that only stage IV-A was significantly different from stage V (p = 0.006) and not stage IV-A vs stage IV-B (p = 0.45) nor stage IV-B vs stage V (p = 0.045).

digitally high IOP. Among these 33 eyes with digitally high IOP, all eyes had shallow AC, 6 eyes had increased corneal diameter (4 of these eyes had Habb's striae) and 23 eyes had corneal edema. Table 3 Shows proportion of eyes with glaucoma, types of glaucoma interventions and surgeries were compared in all stages of ROP.

The mechanism of glaucoma was secondary open angle glaucoma in 35 (42.7%) eyes (3, 4, 12 and 16 in stages II and III, IV-A, IV-B and V respectively), secondary angle closure glaucoma in 39 (47.6%) eyes (0, 3, 7 and 29 in stages II and III, IV-A, IV-B and V respectively) and neovascular in 8 eyes (9.8%). The proportion of eyes with NVG were significantly more (p = 0.018) in IV-A stage of ROP compared to stage V due to active neovascular component in stage 1V ROP compared to stage V.

**Table 3. Types of glaucoma and their interventions in these eyes with ROP and glaucoma.**

| Glaucoma | | II and III* (n = 3 eyes; 3 patients) | IV-A (n = 11 eyes; 8 patients) | IV-B (n = 20 eyes; 13 patients) | V (n = 48 eyes; 34 patients) | p-value |
|---|---|---|---|---|---|---|
| **Neovascular glaucoma** | | 0 (0%) | 4 (36.4%) | 1 (5%) | 3 (6.3%) | **0.008[a]** |
| **Secondary glaucoma (open angle or angle closure)** | | 3 (100%) | 7 (63.6%) | 19 (95%) | 45 (93.7%) | |
| **Age at glaucoma diagnosis (months), median (IQR)** | | 54.9 (6.2 to 54.9) | 5.5 (3.7 to 65.6) | 5.6 (3.4 to 19) | 9.5 (5.3 to 24.9) | 0.57 |
| **Number of glaucoma surgeries (n = 16)** | | 1 (33.3%) | 2 (18.2%) | 5 (25%) | 8 (16.7%) | 0.73 |
| **Type of glaucoma surgery** | AGV | 0 | 0 | 1 | 0 | 0.13 |
| | TSCPC | 0 | 1 | 2 | 8 | |
| | CTT | 1 | 1 | 2 | 0 | |

*Not included in analysis because of small sample size; AGV: Ahmed glaucoma valve; CTT: combined trabeculotomy with trabeculectomy; IQR: inter-quartile range; TSCPC: transscleral cyclophotocoagulation;

[a]Chi-square test; Post-hoc analyses (multiple comparisons by Fisher Exact test) showed that only stage IV-A was significantly different from stage V (p = 0.018) and not stage IV-A vs stage IV-B (p = 0.042) nor stage IV-B vs stage V (p = 1.00).

Disc assessment in these eyes is a challenge due to corneal haze, small pupil and retinal detachment. Disc evaluation was possible in 9 eyes; of these, 4 eyes had 0.5 cup-to-disc ratio (CDR) or more and the other 5 eyes had CDR less than 0.4.

All children were started on topical antiglaucoma medications (AGM) and the medications were adjusted based on the clinical course with treatment. Surgical intervention for IOP control was needed in 16 of the 82 eyes (19.5%). Transscleral cyclophotocoagulation (TSCPC) was performed in 11 eyes, 4 eyes underwent combined trabeculectomy with trabeculotomy (CTT) and 1 eye underwent Ahmed glaucoma valve (AGV) implantation. One eye with stage V ROP needed repeat TSCPC as a second intervention. The median time from glaucoma diagnosis to surgery was 6.9 months (IQR, 4.3–10.8 months). At last follow up, 56 eyes (68%) were on topical antiglaucoma medications, 39 eyes (47%) required single medication, 13 eyes (16%) required 2 medications, 4 eyes (5%) required 3 medications and 26 eyes (32%) did not need any AGM following intervention. The mean number (± standard deviation) of AGM at last follow up was 0.9 ± 0.88. The mean IOP at last visit was 16.0 ± 8.1 mm Hg. The number of antiglaucoma medications (p = 0.49) and IOP at final follow up (p = 0.81) were comparable.

In the eyes that underwent CTT, intra operative limited supra choroidal haemorrhage (SCH) was noted in one eye, pre-retinal and vitreous haemorrhage (VH) was noted in one eye. The complications resolved with conservative management in all 3 eyes. Ahmed glaucoma valve implantation was performed in 1 eye which had Stage IV-B ROP. This eye required one medication post operatively and the IOP was well controlled.

**Visual acuity and refractive error.** The details of vision and refraction of the study cohort is shown in Table 4. In 14 eyes, Snellen or Tellar acuity assessment was possible, in 16 eyes, vision could not be quantified, but had ambulatory vision. 32 eyes had poor/non ambulatory vision, 10 eyes had light perception and 14 eyes had no light perception and in 20 eyes vision could not be assessed. The visual acuity was 20/20 in 2 eyes, 20/30 in 1 eye, 20/200 or better in 4 eyes, between 20/200 and 20/600 in 3 eyes and 20/1400 to 20/3000 in 4 eyes. The mean Log MAR visual acuity in these 14 eyes was 1.14 ± 0.75. Visual acuity was 20/20 in eyes with (one eye each) stage II and stage III ROP, one eye each with stage IVA had 20/30 and 20/60 vision. In summary, in our cohort with ROP and glaucoma, 37% (30 eyes) had ambulatory vision and the rest 63% had poor/non ambulatory vision or vision assessment was not possible.

**Glaucoma in eyes with vitreoretinal surgery.** Fifty-nine eyes of 41 children in this cohort underwent VR surgery. The median time to glaucoma diagnosis from VR surgery was 43.5 days (IQR, 5–154 days) and range was 1 to 3433 days. In 21 eyes, glaucoma was diagnosed in ≤1 month (early glaucoma) and, in 25 eyes, it was diagnosed >1 month (delayed glaucoma). In the early glaucoma group, 17 of the 21 eyes (80.9%) were controlled medically and 4 eyes

**Table 4. Showing details of vision and refraction of the study cohort.**

| Feature | | II and III* (n = 3) | IV-A (n = 11) | IV-B (n = 20) | V (n = 48) | p-value |
|---|---|---|---|---|---|---|
| | **Refraction (D), median (IQR)** | +0.75 (-16 to +0.75) | +9 (+1 to +18) | +13 (+9 to +20) | +16 (+12 to +16.5) | 0.85 |
| **Final vision** | **N** | 2 | 10 | 18 | 32 | |
| **AM** | **Can hold objects, FFL or LogMAR recorded** | 2 (100%) | 7 (70%) | 10 (55.6%) | 11 (34.4%) | 0.10 |
| **NA** | **Light reflex, NPL or PL/PR** | 0 (0%) | 3 (30%) | 8 (44.4%) | 21 (65.6%) | |
| **VA in LogMAR** | | (n = 2) 0.00 and 0.00 | (n = 6) 1.34 ± 0.81 | (n = 5) 1.35 ± 0.42 | (n = 1) 1.10 | N/A |

*Not included in analysis because of small sample size, AM: Ambulatory vision, NA: Non-ambulatory vision, D: Diopters, IQR: Interquartile range, FFL: Fixing following light, NPL: Nil perception of light, PL/PR: perception of light or projection of rays, VA: visual acuity, Log MAR: Logarithm of the minimum angle of resolution. Note: Vision was checked in 62 eyes and the rest 20 eyes vision could not be recorded. 30 eyes (37%) had ambulatory vision, 32 eyes (39%) non-ambulatory vision, 20 (24%) eyes was not possible to assess.

(19.1%) had persistent IOP elevation and needed glaucoma surgery. The median duration to glaucoma surgery was 8.8 months (IQR, 6.7 to 10.8 months). In delayed glaucoma group, 19 of the 25 eyes (76%) were treated medically and glaucoma surgery was needed in 6 eyes (24%) (p = 0.74, Fisher Exact test). The median duration to glaucoma surgery was 9.8 months, (IQR, 6.9 to 27.7 months). Although we could not find significant relationship between stage of ROP and time to development of glaucoma after VR surgery (p = 0.13, Kruskal-Wallis test), glaucoma developed much earlier in eyes with stage V ROP. The median duration for development of glaucoma in eyes with Stage V ROP (n = 22) was 10 days (IQR, 1 to 125 days) whereas for eyes with Stage IVA (n = 7) was 88 days (IQR, 48 to 103 days) and Stage IVB (n = 17) was 95 days (IQR, 31 to 154 days).

## Discussion

The Indian Twin cities ROP Screening (ITCROPS) is a long-term prospectively collected data of all children at a single centre using uniform methodology of assessment with a single team leader (SJ) since 1997.

In our study, 1.36% of eyes with ROP had secondary glaucoma and majority of them were eyes with stage V (59%) and IV (37.8%) ROP. Secondary glaucoma in retinopathy of prematurity is not uncommon. It was estimated that 30% of the children with untreated stage V ROP develop secondary glaucoma [14] In the ETROP study, among the children treated for high risk pre-threshold ROP, 1.67% children developed glaucoma during first 6 years of life [13].

Several mechanisms have been described for the development of glaucoma in eyes with ROP, commonest being angle closure [7, 9, 12, 19]. Many of these studies have diagnosed angle closure based on shallow anterior chamber or presence of peripheral anterior synechiae. In the ETROP study that described glaucoma in 12 eyes with ROP, the majority (7/12, 58.3%) had shallow anterior chamber [13]. In a series by Chandra et al, all 6 eyes with glaucoma after VR surgery for stage IV ROP had open angles and the authors had proposed angle dysgenesis as the possible mechanism for glaucoma [6]. In our series, the shallow anterior chamber or closed angles were noted in 47% eyes (39/82) and deep anterior chamber or open angle in 43% eyes (35/82 eyes). There was no difference in the outcomes of these eyes with regards to IOP control. Several of these studies including ours are limited by the lack of gonioscopy findings at the time of diagnosis and follow up in these premature babies.

Surgical intervention for IOP control was required in close to 20% of them and 68% of the eyes were on topical AGM at the last follow up. There are very few studies till date describing types of glaucoma and their management in eyes with various stages of ROP. The details of glaucoma and their management in various studies are shown in Table 5. None of the studies have mentioned any intraoperative complications in eyes that underwent glaucoma surgery. We noted vitreous haemorrhage, subhyaloid haemorrhage and limited suprachoroidal haemorrhage in one eye each following CTT. Intraocular bleed in these eyes can occur following any intervention due to abnormal peripheral retinal vascularity. Sudden decompression during glaucoma surgery may predispose the eyes to bleed. It is also possible that we may have noted these complications since postoperative ultrasound B scan was performed as a routine at 1 day and 1 week. Although the complications completely resolved with conservative management, awareness about this complication is important and precautions have to be taken to prevent them or identify early and treat appropriately.

As was seen on our series, the proportion of eyes with neovascular glaucoma were significantly more in stage IVA ROP compared to stage V. This is due to higher levels of neovascular cytokines like VEGF which are significantly more in eyes with stage IVA ROP [20]. Hence during vascularly active phase of ROP, neovascular glaucoma should be looked for and treated

**Table 5. Table showing the studies that reported glaucoma in ROP and their interventions.**

| Sno | Author (year) | number of eyes with ROP | Follow up | No of eyes with Glaucoma | No of eyes with Open angle | No of eyes with Angle closure | Medical treatment | Glaucoma surgery |
|---|---|---|---|---|---|---|---|---|
| 1 | Hartnett (1993) | 26 | 2 years | 7 | 3 | 4 | 5 | 2 Cyclocryo |
| 2 | Bremer (2012) | 718 | 6 years | 12[$] | 4 | 7 | 1 | 3 (2 LS, 1 AGV) |
| 3 | Iwahashi-Shima (2012) | 55[#] | 2 years | 8 | 5 | 2 | 2 | 6 (4 TLO, 1Trab, 1 GSL) |
| 4 | Chandra (2019) | 100[*] | 1.25 years | 6 | 6 | | 2 | 4 (CTT with MMC) |
| 5 | Senthil (current study) | 6000 | 0–12.6 years | 82 | 35 | 39 | 66 | 16 (4 CTT, 1 AGV, 11 TSCPC) |

[#] Post VR surgery for stage 4 and 5 ROP;

[*] These were post VR surgery for stage 4 ROP;

[$] only 4 eyes were treated, 8 eyes with RD were not treated for glaucoma due to poor prognosis; LS: lensectomy, TLO: trabeculotomy, CTT: combined trabeculotomy and trabeculectomy, GSL: goniosynechiolysis, TSCPC: transscleral cyclophotocoagulation, AGV: Ahmed glaucoma valve.

appropriately with retinal ablation and or antiVEGF injections as the clinical picture dictates. We recommend close monitoring during follow up for early detection of this complication in eyes with stage IV ROP.

In the earlier publication by Iwahashi-shima et al, young gestational age (OR:1.14) and lensectomy (OR:8.79) were significant predictors of delayed-onset IOP elevation [5]. Vitrectomy increased the risk of IOP elevation [4], lensectomy along with vitrectomy increased the risk of IOP elevation even higher (22.6% vs 4.2%) [5]. These findings was reversed in a recent study by Chandra et al., who found delayed glaucoma in eyes with combined lensectomy and vitrectomy compared to those with lens sparing vitrectomy [6]. We did not find such association or predictors in our cohort. Also in our cohort, the effect of lensectomy on the development of glaucoma could not be assessed since close to 91% of the eyes that underwent vitreoretinal surgery had simultaneous lensectomy as well. This is due to large numbers of stage V ROP referred and operated at our centre.

In our cohort, 39% had ambulatory vision and the rest 61% had poor/ unrecordable vision. Glaucoma is known to be a cause of poor vision in advanced ROP even after successful retinal outcome [4]. Although a single case study, Harnett et al have demonstrated that reducing the intraocular pressure helps to improve vision in these children with ROP and glaucoma [21]. Hence, close monitoring for glaucoma in these children even after successful retinal intervention for ROP is recommended to improve visual outcomes. Childhood glaucoma affects vision due to corneal edema/scar and or optic disc damage. When children with ROP have associated glaucoma, the visual outcomes could be poor due to two serious sight threatening conditions. Our data throws some light on the visual outcomes in children with ROP complicated with glaucoma. Overall, 39% of eyes had ambulatory vision in our series, and we noted better visual outcomes in eyes with less severe ROP compared to those with advanced stages of ROP. It would be important to closely monitor these children with a multidisciplinary approach with protocols of vision assessment, refraction, glaucoma evaluation and retinal examination at every visit, besides close collaboration with early intervention specialists, low vision and rehabilitation specialists [15, 18].

Our study, though is the single largest series on glaucoma in ROP, has its limitations. One of the limitations of our study is that the diagnosis of open angle or angle closure glaucoma was made based on the clinical assessment of anterior chamber depth by the glaucoma specialist and not based on the gonioscopic findings. This may have bias in the classification of the subtypes of glaucoma and may have wrongly classified a few of the eyes. Previous studies have

described angle closure to be the cause of glaucoma in children with shallow anterior chamber depth without gonioscopy [12, 13]. Another limitation is non availability of visual acuity data for most of the eyes. Difficulty in assessing visual acuity in these children is compounded by the lack of regular follow up specially if the vision is poor in these children. In some of the children, finger tension was used to assess IOP when general anaesthesia (GA) could not be administered for medical reasons. Considering the difficulty in subjecting them to repeated general anaesthesia, finger tension IOP estimation during follow up is a limitation however cannot be avoided in these children. In the recent times, rebound tonometry is of great use in assessing IOPs in the clinic in children without subjecting them to GA.

In conclusion, glaucoma is an important cause of ocular morbidity in children with advanced stage IV and V ROP. Early diagnosis and management in these cases is a challenge considering the difficulties in evaluation, diagnosis and management. It is important to create awareness among the retina specialists about glaucoma as a serious complication in ROP eyes following a vitreoretinal intervention. It is also important to counsel the parents regarding the complexity of the problem and continuity of care in children with ROP and glaucoma.

## Acknowledgments

**Presentation at a meeting**: World Congress for Pediatric Ophthalmology and Strabismus in December 2017 and Indian Pediatric glaucoma society meeting in December 2019.

## Author Contributions

**Conceptualization:** Sirisha Senthil, Subhadra Jalali.

**Data curation:** Sirisha Senthil, Pasyanthi Balijepalli, Ashik Mohamed.

**Formal analysis:** Sirisha Senthil, Ashik Mohamed.

**Methodology:** Sirisha Senthil.

**Supervision:** Sirisha Senthil.

**Validation:** Sirisha Senthil, Chandrasekar Garudadri.

**Writing – original draft:** Sirisha Senthil, Pasyanthi Balijepalli.

**Writing – review & editing:** Sirisha Senthil, Ashik Mohamed, Padmaja Kumari Rani, Sameera Nayak, Anil K. Mandal, Subhadra Jalali.

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
