## [Decision Letter · Decision Letter 0]

11 Jun 2020

PONE-D-20-14601

Management outcomes of secondary glaucoma due to Retinopathy of Prematurity: a 19-year prospective study at a tertiary eye care Institute. The Indian Twin cities ROP Screening (IRCROPS) database report number 8

PLOS ONE

Dear Dr. Senthil,

Thank you for submitting your manuscript to PLOS ONE. After careful consideration, we feel that it has merit but does not fully meet PLOS ONE’s publication criteria as it currently stands. Therefore, we invite you to submit a revised version of the manuscript that addresses the points raised during the review process.

This is an important study and both reviewers agree. However, there are som issues with the statistical analysis and one of the expert reviewers cannot replicate the statistical values calculated. We will please ask you to look carefully at this statistics provide a detailed explanation and include a statistician in the revision if one is nto already included.  We look forward to the revised submission. 

We look forward to receiving your revised manuscript.

Kind regards,

Demetrios G. Vavvas

Academic Editor

PLOS ONE

Journal Requirements:

Reviewers' comments:

Reviewer's Responses to Questions

**Comments to the Author**

1. Is the manuscript technically sound, and do the data support the conclusions?

Reviewer #1: Yes

Reviewer #2: Partly

2. Has the statistical analysis been performed appropriately and rigorously? 

Reviewer #1: Yes

Reviewer #2: No

3. Have the authors made all data underlying the findings in their manuscript fully available?

Reviewer #1: Yes

Reviewer #2: No

4. Is the manuscript presented in an intelligible fashion and written in standard English?

Reviewer #1: Yes

Reviewer #2: Yes

5. Review Comments to the Author

Reviewer #1: The authors report the largest cohort to date of glaucoma secondary to ROP in a comprehensive and thorough fashion. Statistical analysis is performed in a sound way and results support this study's conclusions.

They are invited to consider the following few comments to strengthen their work:

1. Out of 82 eyes included in this study, only 14 had VA assessment in the final follow-up. Please add mean follow-up for the whole cohort in terms of VA assessment. This would help the Editor and potential reading audience understand to what extent the lack of ‘final VA’ in the vast majority of this cohort weakens the secondary outcome of this study

2. 34 of of 54 eyes included in this study were stage V which explains the relatively high rates of secondary glaucoma and vitreoretinal surgery in this cohort compared to the literature.

Minor comments

3. Abstract lines 32-33 should read: Eighty-two eyes of 54 children with secondary glaucoma due to ROP where included in this study

4. Line 60 should read: ‘These include’ instead of ‘they are’

5. Line 140 PCG should be explained when first used

Reviewer #2: In the paper by Senthil et al, they report a large cohort of premature babies diagnosed with ROP, a subset of whom who were found to have concurrent glaucoma. There are two aspects to this paper – the first is the reporting of the prevalence of glaucoma and types of interventions in patients with ROP, and the second is estimating correlations between ophthalmic findings or interventions and stage of ROP in patients with glaucoma. The size of the cohort and the manner in which patients were collected is sound and robust, and though this is not the first paper to review glaucoma in ROP patients (Chandra et al., for example), it is the largest and verifies previous findings. It also reiterates that glaucoma prevalence goes up as ROP stage worsens, and one must not miss this diagnosis when evaluating patients with ROP. It is important to publish such data. I have concerns about the statistical approach for the second aspect of the paper, which is reporting correlations between ophthalmic findings or type of glaucoma and stage of ROP.

While I believe the authors have strong data with good information to report, the representation of the data and statistical approach is concerning and should be revised on multiple levels.

1. I cannot replicate the chi-squared results. In Table 1, I assume this may be due to the way patients and eyes are treated, and so I will skip this table. I cannot replicate the exact p-values in Table 2. For example – Laser indirect ophthalmoscopy, category “0”. If I ignore the first column as the authors say they did (due to sample size), I get a p-value of 0.0002 for a 2x3 chi-squared contingency table where the rows are “0” and “not 0” and the columns are stage of ROP. The authors report a p-value of 0.50. I am having this same problem for some – if not all – of the data. The authors need to clarify significantly exactly what test they are doing to yield these results and what is the denominator. But there are further concerns about the approach as outlined below. Of note, I can replicate the subsequent post-hoc pairwise chi-squared tests that were done to identify the source of a signal.

2. For the continuous variables, Kruskal-wallis (and subsequently individual Mann-Whitneys) are used, and this is appropriate.

3. For the other variables that have been treated as categorical, it gets a bit more nuanced. In general, it is misleading to calculate a separate chi-square for each subcategory within a category (for example – treating number of lasers 0,1,2 as three categories and calculating separate statistics). Within a category, these tests are not independent – if an eye takes on one subcategory, it is by definition not the other. Rather, one can do a multi-way chi-squared test on the full 3x3 (or 2x3 or 4x3, depending on the category) contingency table, and then report the pairwise comparisons therein that likely represent the signal, as the authors have done in the Table caption. This results in one p-value per category.

4. A Fisher’s test should have been used rather than a chi-squared if any category has 10 or fewer total counts.

5. It is sub-optimal to use the chi-squared test on ordinal categorical data. I would suggest a Kruskal-Wallis for these data, but there are other options the authors could choose as well. For example – the number of lasers (0,1,2) is converted into 3 independent categories; these are ordered and not unrelated, so they should be treated as such.

6. The Bonferonni correction is unusual. The subsequent pairs of mann-whitneys (or chi-squares) after the multi-way Kruskal-wallis (or chis-square) are not at all independent from each other, and therefore it doesn’t make sense to correct with Bonferonni (and correcting for 2 tests when more than 2 were done is also incorrect). If the authors want to correct for multiple testing, consider a Dunn’s post-hoc correction in this setting after a Kruskal-Wallis (or consult with their biostatistician on the best way to do this). In addition, the real Bonferonni correction that should have been done is to correct for the number of categories that were tested, as these are all independent of one another (for example, in Table 2, correct for 5 tests -- # of VR surgeries, age, laser, avastin, lensectomy).

These problems make it somewhat hard and confusing to read the paper and glean the important takeaways.

Finally, one other question for the authors is why they are calculating statistics within the glaucoma group and not comparing glaucoma to non-glaucoma? For example – the number of Avastin injections in patients who developed or didn’t develop glaucoma? They have a wealth of data and these seem like the important questions.

I do believe that overall, the trend of their results will be similar with the above corrections, but because of these concerns, it was hard to dive deeper into interpreting the results. They should consult with someone who has training in statistics in order to clean up the results and form robust conclusions.

6. PLOS authors have the option to publish the peer review history of their article (what does this mean?). If published, this will include your full peer review and any attached files.

Reviewer #1: No

Reviewer #2: No

---

## [Author Response · Author response to Decision Letter 0]

14 Jul 2020

Response to Reviewers Comments 

Reviewer #1: The authors report the largest cohort to date of glaucoma secondary to ROP in a comprehensive and thorough fashion. Statistical analysis is performed in a sound way and results support this study's conclusions.

Response: We that the reviewer for the compliment and encouraging words.

They are invited to consider the following few comments to strengthen their work:

1. Out of 82 eyes included in this study, only 14 had VA assessment in the final follow-up. Please add mean follow-up for the whole cohort in terms of VA assessment. This would help the Editor and potential reading audience understand to what extent the lack of ‘final VA’ in the vast majority of this cohort weakens the secondary outcome of this study

Response: Thank you for the suggestion. We have provided the details of mean follow up for the visual acuity assessment in the abstract as well as in the results. Page 2, lines 41,42, page 7, lines 149, 150.

Mean follow up was 3.5±4.2 years in those whose Log MAR visual acuity was recorded. The mean follow up for the entire cohort was 1.14±2.24 years.

2. 34 of 54 eyes included in this study were stage V which explains the relatively high rates of secondary glaucoma and vitreoretinal surgery in this cohort compared to the literature.

Response: Yes, we agree that the incidence of glaucoma is high in eyes with stage V ROP that are untreated, however, it seems higher even post VR surgery.

Minor comments

3. Abstract lines 32-33 should read: Eighty-two eyes of 54 children with secondary glaucoma due to ROP where included in this study

Response: We have changed this as suggested. Page 2, line 33.

4. Line 60 should read: ‘These include’ instead of ‘they are’

Response: We have changed as suggested. Page 3, line 62.

5. Line 140 PCG should be explained when first used

Response: Sorry about this. We have expanded the abbreviation. Page 6, line 145.

Reviewer #2: 

In the paper by Senthil et al, they report a large cohort of premature babies diagnosed with ROP, a subset of whom who were found to have concurrent glaucoma. There are two aspects to this paper – the first is the reporting of the prevalence of glaucoma and types of interventions in patients with ROP, and the second is estimating correlations between ophthalmic findings or interventions and stage of ROP in patients with glaucoma. The size of the cohort and the manner in which patients were collected is sound and robust, and though this is not the first paper to review glaucoma in ROP patients (Chandra et al., for example), it is the largest and verifies previous findings. It also reiterates that glaucoma prevalence goes up as ROP stage worsens, and one must not miss this diagnosis when evaluating patients with ROP. It is important to publish such data. I have concerns about the statistical approach for the second aspect of the paper, which is reporting correlations between ophthalmic findings or type of glaucoma and stage of ROP.

While I believe the authors have strong data with good information to report, the representation of the data and statistical approach is concerning and should be revised on multiple levels.

Response: As suggested by the reviewer, we have revised the data representation and statistical analysis.

1. I cannot replicate the chi-squared results. In Table 1, I assume this may be due to the way patients and eyes are treated, and so I will skip this table. I cannot replicate the exact p-values in Table 2. For example – Laser indirect ophthalmoscopy, category “0”. If I ignore the first column as the authors say they did (due to sample size), I get a p-value of 0.0002 for a 2x3 chi-squared contingency table where the rows are “0” and “not 0” and the columns are stage of ROP. The authors report a p-value of 0.50. I am having this same problem for some – if not all – of the data. The authors need to clarify significantly exactly what test they are doing to yield these results and what is the denominator. But there are further concerns about the approach as outlined below. Of note, I can replicate the subsequent post-hoc pairwise chi-squared tests that were done to identify the source of a signal.

Response: The tables were modified and should reflect the correct values now. Tables 1-4, and the statistical analysis.

2. For the continuous variables, Kruskal-Wallis (and subsequently individual Mann-Whitneys) are used, and this is appropriate.

Response: We thank the reviewer for the comment.

3. For the other variables that have been treated as categorical, it gets a bit more nuanced. In general, it is misleading to calculate a separate chi-square for each subcategory within a category (for example – treating number of lasers 0,1,2 as three categories and calculating separate statistics). Within a category, these tests are not independent – if an eye takes on one subcategory, it is by definition not the other. Rather, one can do a multi-way chi-squared test on the full 3x3 (or 2x3 or 4x3, depending on the category) contingency table, and then report the pairwise comparisons therein that likely represent the signal, as the authors have done in the Table caption. This results in one p-value per category.

Response: We understand the concern. We have simplified to reflect them better. Analysis were done for 2x3 tables now resulting in one p-value per category.

4. A Fisher’s test should have been used rather than a chi-squared if any category has 10 or fewer total counts.

Response: As suggested by the reviewer, we have used Fisher Exact test for multiple pair-wise comparisons now.

5. It is sub-optimal to use the chi-squared test on ordinal categorical data. I would suggest a Kruskal-Wallis for these data, but there are other options the authors could choose as well. For example – the number of lasers (0,1,2) is converted into 3 independent categories; these are ordered and not unrelated, so they should be treated as such.

Response: The tables were modified and the ordinal categorical data were eliminated in order to avoid confusion.

6. The Bonferonni correction is unusual. The subsequent pairs of mann-whitneys (or chi-squares) after the multi-way Kruskal-wallis (or chis-square) are not at all independent from each other, and therefore it doesn’t make sense to correct with Bonferonni (and correcting for 2 tests when more than 2 were done is also incorrect). If the authors want to correct for multiple testing, consider a Dunn’s post-hoc correction in this setting after a Kruskal-Wallis (or consult with their biostatistician on the best way to do this). In addition, the real Bonferonni correction that should have been done is to correct for the number of categories that were tested, as these are all independent of one another (for example, in Table 2, correct for 5 tests -- # of VR surgeries, age, laser, avastin, lensectomy).

Response: After consulting with our bio-statistician, we have represented the data in a simpler way. The tables were revised and appropriate tests were applied now considering the previous comments of Reviewer #2.

These problems make it somewhat hard and confusing to read the paper and glean the important takeaways.

Response: We have worked towards eliminating these problems during revision in order to avoid confusion reading the paper.

Finally, one other question for the authors is why they are calculating statistics within the glaucoma group and not comparing glaucoma to non-glaucoma? For example – the number of Avastin injections in patients who developed or didn’t develop glaucoma? They have a wealth of data and these seem like the important questions.

Response: We thank the reviewer for the suggestion. We agree with the suggestion, however it is beyond the scope of this manuscript.

I do believe that overall, the trend of their results will be similar with the above corrections, but because of these concerns, it was hard to dive deeper into interpreting the results. They should consult with someone who has training in statistics in order to clean up the results and form robust conclusions.

Response: We have re-evaluated the results and made necessary corrections. We agree that the results did not change however there was a small variation in the numbers. Hope the analysis and results are fine now.

---

## [Decision Letter · Decision Letter 1]

21 Aug 2020

Management outcomes of secondary glaucoma due to Retinopathy of Prematurity: a 19-year prospective study at a tertiary eye care Institute. The Indian Twin cities ROP Screening (IRCROPS) database report number 8

PONE-D-20-14601R1

Dear Dr. Senthil,

We’re pleased to inform you that your manuscript has been judged scientifically suitable for publication and will be formally accepted for publication once it meets all outstanding technical requirements.

Kind regards,

Demetrios G. Vavvas

Academic Editor

PLOS ONE

Additional Editor Comments (optional):

Reviewers' comments:

Reviewer's Responses to Questions

**Comments to the Author**

1. If the authors have adequately addressed your comments raised in a previous round of review and you feel that this manuscript is now acceptable for publication, you may indicate that here to bypass the “Comments to the Author” section, enter your conflict of interest statement in the “Confidential to Editor” section, and submit your "Accept" recommendation.

Reviewer #1: All comments have been addressed

Reviewer #2: All comments have been addressed

2. Is the manuscript technically sound, and do the data support the conclusions?

Reviewer #1: Yes

Reviewer #2: Yes

3. Has the statistical analysis been performed appropriately and rigorously? 

Reviewer #1: Yes

Reviewer #2: Yes

4. Have the authors made all data underlying the findings in their manuscript fully available?

Reviewer #1: No

Reviewer #2: Yes

5. Is the manuscript presented in an intelligible fashion and written in standard English?

Reviewer #1: Yes

Reviewer #2: Yes

6. Review Comments to the Author

Reviewer #1: No further comments, comments raised in the previous round of review have been adequately addressed in the revised version

Reviewer #2: The authors have responded to my concerns. The p-values appear to be correct now, and the analyses are sound. As suspected, the conclusions are similar.

I still think it is redundant to have separate rows for "male" and female" or "yes surgery" "no surgery" as these are mutually exclusive categories, but this is a minor comment and more of a stylistic thing to de-clutter the tables. You just need "% male" - you don't need another row with %female, as it is by definition 100-%male. Same with yes/no categories.

7. PLOS authors have the option to publish the peer review history of their article (what does this mean?). If published, this will include your full peer review and any attached files.

Reviewer #1: No

Reviewer #2: No

---

## [Editor Report · Acceptance letter]

27 Aug 2020

PONE-D-20-14601R1 

Management outcomes of secondary glaucoma due to Retinopathy of Prematurity: a 19-year prospective study at a tertiary eye care Institute. The Indian Twin cities ROP Screening (ITCROPS) database report number 8 

Dear Dr. Senthil:

I'm pleased to inform you that your manuscript has been deemed suitable for publication in PLOS ONE. Congratulations! Your manuscript is now with our production department. 

Kind regards, 

on behalf of

Dr. Demetrios G. Vavvas 

Academic Editor

PLOS ONE